# Internal Waves Study on a Narrow Steep Shelf of the Black Sea Using the Spatial Antenna of Line Temperature Sensors

**Andrey Serebryany** [1,2,*], **Elizaveta Khimchenko** [1], **Oleg Popov** [3], **Dmitriy Denisov** [2] **and Genrikh Kenigsberger** [4]

[1] Shirshov Institute of Oceanology, Russian Academy of Sciences, 117997 Moscow, Russia; ekhymchenko@gmail.com

[2] Andreyev Acoustics Institute, 117036 Moscow, Russia; denisov.dimitriy@gmail.com

[3] Obukhov Institute of Atmospheric Physics, Russian Academy of Sciences, 119017 Moscow, Russia; olegp@mail.ru

[4] Institute of Ecology of the Academy of Sciences of Abkhazia, Sukhum 384900, Abkhazia; kenigsbergerg@mail.ru

\* Correspondence: serebryany@hotmail.com

**Abstract:** The results of investigations into internal waves on a narrow steep shelf of the northeastern coast of the Black Sea are presented here. To measure the parameters of internal waves, the spatial antenna of three autonomous line temperature sensors were equipped in the depth range of 17 to 27 m. In observations that lasted for 10 days, near-inertial internal waves with a period close to 17 h and short-period internal waves with periods of 2–8 min, regularly approaching the coast, were revealed. The wave amplitudes were 4–8 m for inertial waves and 0.5–4 m for short-period internal waves. It was determined that most of the short-period internal waves approached from the southeast direction, from Cape Kodor. A large number of short waves reflected from the coast were also recorded. The intensification of short-period waves with inertial periodicity and the belonging of trains of short waves to crests of inertial waves were identified. In general, it was shown that the internal wave field at a narrow shelf significantly differs in its features from analogs of ordinary shelves of the Black Sea.

**Keywords:** internal waves; inertial and short-period internal waves; space antenna; line temperature sensors; shelf; the Black Sea

## 1. Introduction

The Black Sea is a unique phenomenon from many points of view, but here we will focus on only one aspect: internal waves. Internal waves are widespread all over the world's oceans, including in marginal and inland seas. Internal waves propagate in a vertically stratified medium and perform a significant function as the main source of internal mixing (internal ventilation) in the ocean. The base source of the generation of internal waves is tidal forces. A large number of works have been devoted to field studies of internal waves in oceans and seas around the world [1–5]. Initially, investigations were carried out using contact methods. In recent decades, remote sensing methods have also been used [6,7]. Large-scale experiments have been conducted on ocean shelves, where the high intensification of internal waves are found [8–11]. As a result, there is information on the main characteristics of internal waves in various regions of the ocean, and regions of the world's oceans where large-amplitude internal waves are known to exist. The features of internal waves, as nonlinear waves exhibiting the properties of solitons, have been described. In recent years, evidence was found of the existence of second mode internal waves in the ocean [12,13]. Various important effects associated with internal waves have

also been studied [14,15], leading to progress in the investigation of internal wave generation, among which, tidal mechanisms are the best studied [16,17].

In contrast to the ocean and open seas, in the enclosed Black Sea, tides are small [18] and largely do not affect the formation of the field of internal waves. However, internal waves do exist here, playing an important role in vertical mixing of the water column. Observations of internal waves in the Black Sea cover a period of several decades [19–25]. In the specialized observations that have been carried out, not only have the parameters of internal waves been measured, but also the mechanisms of their generation have been identified [26–28]. The internal wave field can be divided into two ranges, one of which is long internal waves with an inertial period. In the Black Sea, this has a range of about 17 h. The range of short-period internal waves has a frequency close to the buoyancy frequency. The periods of these waves are from minutes to several tens of minutes. Most of the observations of internal waves in the Black Sea have been carried out in the shelf zones. The properties of the shelf lead to the features of the observed field of internal waves. From this perspective, observations on the steep and narrow shelf in the northeastern part of the Black Sea, where we conducted our investigations, are of great interest. In the Black Sea, inertial internal waves perform the role of the internal tides of the most energy-intensive internal waves. As previous observations have shown, inertial waves on a steep and narrow shelf have larger amplitudes compared to waves on ordinary shelves [29].

## 2. Study Area

In October–November 2019, we conducted a specialized experiment to study internal waves on a narrow shelf near the Caucasian coast of the Black Sea. The measurements were carried out in the coastal zone near Cape Sukhumsky (Figure 1). The shelf of the coast of the cape is narrow and steep, with a sharp drop in depth. The average slope of the bottom at the measurement site, near the platform, is about 23° (Figure 2). The 100 m isobath lies about 200 m from the coast, while on the ordinary shelf, for example, on the northeastern shelf near Gelendzhik, the 100 m isobath is 5 km from the shore.

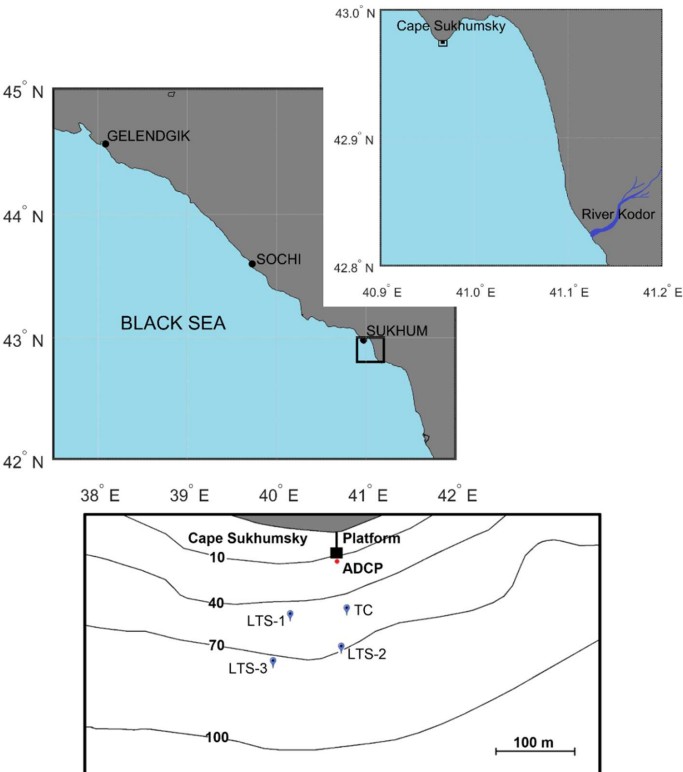

**Figure 1.** The northeast part of the Black Sea shoreline and position of the study site. Map showing mooring locations and bathymetry of the study region (below). Depth is in meters. The dots mark the position of moorings equipped with line temperature sensors (LTSs) and thermistor chain (TC).

In the study area, north-west alongshore currents prevail. Often, there is an increase in the current velocity up to 0.5–1 m/s due to the rim current approaching the coast. This circumstance leads to an increase in inertial oscillations, even during calm weather.

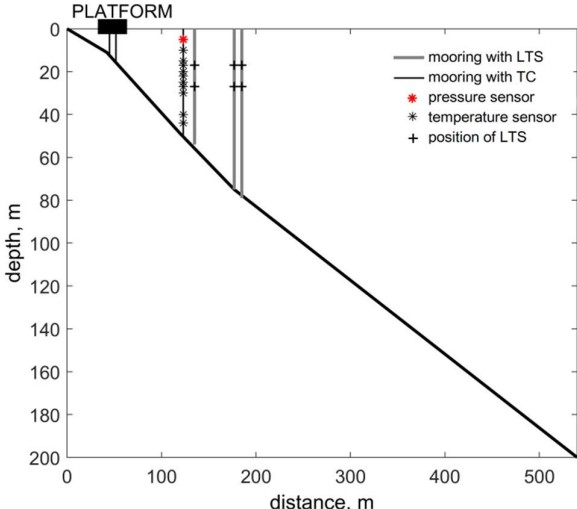

**Figure 2.** Bottom relief at the measurement site, and the position of the measuring instruments.

## 3. Materials and Methods

The originality of this experiment was in using the antenna of line temperature sensors (LTSs), which were employed along with other measuring instruments commonly used in our practice. The LTS is a well-known internal wave meter [30,31] in the form of an insulated wire, which is located vertically in the thermocline and measures the average temperature of the covered water layer, recording the oscillations of internal waves propagating along the thermocline. The main advantage of an LTS over a point sensor is that its recordings are free from distortion, which can be introduced by the fine-structured irregularity of the vertical temperature profile usually found in real marine conditions. Fluctuations of the average temperature measured by an LTS can be easily converted into vertical displacements of the thermocline if the vertical temperature gradient is known, or special calibration is undertaken. The spatial antennas were created based on the LTSs. Previously, with their help, measurements were carried out on the shelves of seas from stationary platforms [26,27,32], as well as in the ocean with towed antennas and antennas deployed while drifting [33].

In our experiment, we used autonomous sensors based on the LTS technical description presented in Denisov and Serebryany [34]. Four moorings holding temperature sensors were installed at a short distance from the coast. The positions of the moorings and their equipment, with instruments against the bathymetry, are shown in Figures 1 and 2. Three stations were equipped with LTSs of 10 m length (№1, №2, №3 in Figure 1), and at the fourth mooring a thermistor chain (TC) with 10 temperature sensors (DST-centi-T of the Star-Oddi) was set. All sensors on the mooring were attached to cables, at one end of which there was a dead anchor, and at the other a submerged buoy. The stations with LTSs were located in the corners of a triangle, with sides of 63 m, 77 m, and 89 m. Line temperature sensors were set at the thermocline at depths ranging from 17 m to 27 m. The sea depths at the mooring locations were 47 m, 48 m, 70 m, and 73 m. The sampling rate of the LTSs was 20 s, and for thermistors was 30 s. To measure the current from the surface to the bottom during the experiment, the acoustic Doppler current profiler (ADCP) "Rio Grande 600 kHz" was installed next to the spatial antenna, from the stationary platform, where the sea depth was 12 m.

Measurements were performed from 24 October until 4 November 2019. The pressure sensor DST-centi-TD was deployed at the thermistor chain at a 5 m depth, to control the vertical position of the mooring, and recorded a stable vertical position of the measuring system for most of the observation

period. A strong current, which caused a deviation of the measuring systems from the upright position, was observed on 3 November. We analyzed the data from 24 October to 2 November 2019.

## 4. Results

The temperature structure of the sea before the observations was characterized by the presence of a pronounced thermocline at depths of 17–35 m. Figure 3 shows the vertical profiles of the temperature and buoyancy frequency at the beginning of measurements, and after six days. In the thermocline, the temperature difference reached 10 °C. The upper quasi-homogeneous layer had a temperature of about 21 °C. LT sensors were placed within the horizons of 17–27 m, as shown in Figure 3. The maximum buoyancy frequency, as seen in Figure 3, reached 30 c/h.

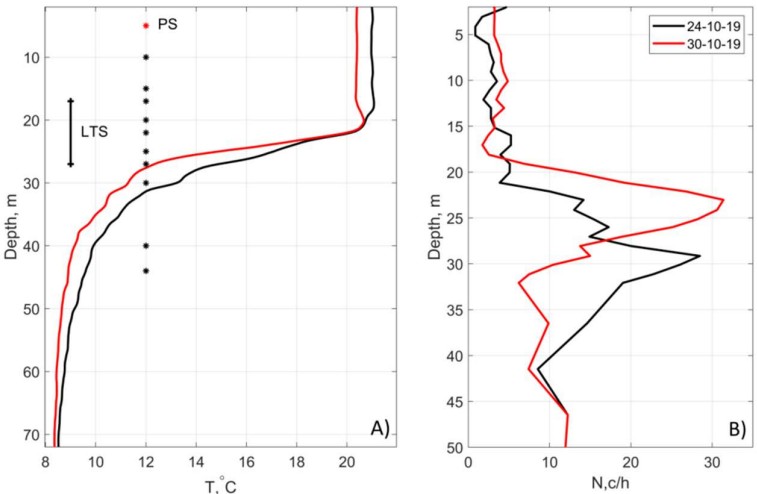

**Figure 3.** Measured vertical temperature profiles and calculated buoyancy frequency profiles from 24 to 30 October 2019. The positions of the LTS and point sensors of the TC are shown. PS: pressure sensor.

The picture of temporal variability of the temperature structure for 10 days of observations was created according to the data of the thermistor chain (Figure 4). Figure 4 shows that the thermocline, on average, tended to shift upward (in 10 days, the lower boundary rose from 30 m to 20 m). At the same time, periodic vertical oscillations of the thermocline, with a period close to the local inertial one of about 17 h, are clearly visible. Over 200 h of observation, there were 12 such oscillations.

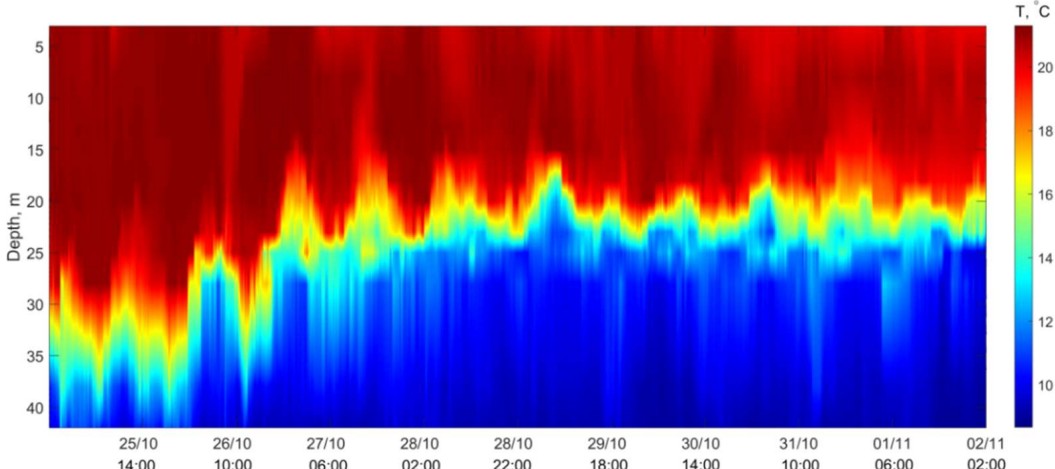

**Figure 4.** Thermocline displacements caused by propagation of inertial internal waves from 25 October to 2 November 2019. Measurements were made by using a temperature sensor chain.

The aforementioned upward movement of the mean position of the thermocline (in Figure 4) was most intense from October 26 and continued for three days. At that moment, according to the ADCP data, the observed alongshore northwestern current first weakened from 0.25 m/s to 0.1 m/s, and then it changed to an intense southeastern current with a velocity up to 0.35 m/s. This restructuring of currents in the coastal zone was the principal cause of the noted upward displacement of the thermocline since the southeastern current in the coastal area of the Black Sea leads to the upward movement of cold waters, which occurs during upwelling. There was no more significant rise in the thermocline since an intense southeastern current has been observed only for 20 h and then a gradual weakening to 0.1 m/s.

The heights of these inertial internal waves varied from 4 m to 8 m. It should be noted that during the entire observation period, the weather conditions were predominantly sunny and calm, with low wind. Besides the long-period oscillations of the thermocline, short-period internal waves with periods of 6–10 min were identified in the record of the thermistor chain.

The information about internal waves, derived by the thermistor chain, was significantly supplemented by the records of the spatial antenna of the line temperature sensors. Figure 5 presents an average frequency spectrum calculated by data recorded on the LTS of station №3. During the calculation of the spectrum, averaging was carried out over the entire series, with an analysis window length of 5 days. Let us recall that the LTS record represents the time series of the temperature of the layer that the LTS covers. In our case, the sensor had a length of 10 m and covered a layer between the horizons of 17 and 27 m. The passing internal waves created vertical displacements of the thermocline, which were reflected in the temperature fluctuations recorded by the line temperature sensor. If we know the vertical gradient of the layer temperature, it is possible to calculate the recorded temperature oscillations into vertical displacements of the thermocline, expressed in meters. Thus, the spectrum calculated from the LTS data is the spectrum of vertical displacements of the water column. However, the dimension of the spectrum shown in Figure 5 is given in relative units. The spectrum reaches the maximum at a frequency of 0.0563 c/h (period 17.8 h), which is close to the local inertial frequency in the study area (17.6 h). Additionally, there is a rise in the spectrum at the high-frequency band (10–20 c/h), which coincides with short-period oscillations of the thermocline. Both peaks are above the calculated 95% confidence interval. Thus, the frequency spectrum, according to the LTS recording data, confirms the presence of inertial and short-period internal waves in the observations.

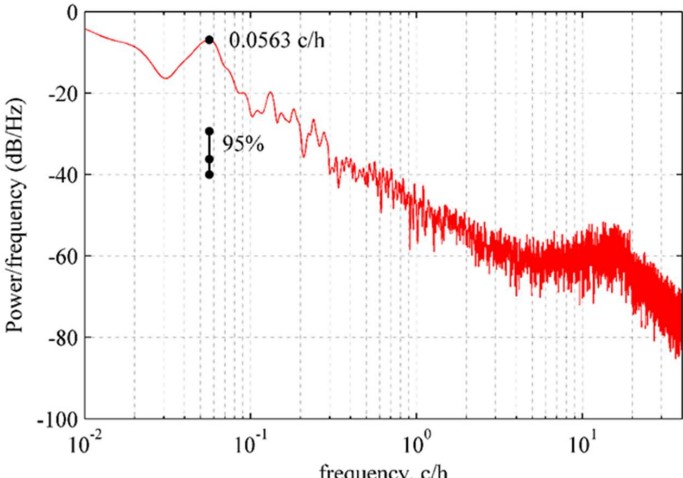

**Figure 5.** The frequency spectrum of internal waves calculated from the temperature record of the LTS (sensor No. 3) over a duration of 10 days. The 95% confidence interval is shown.

To better understand the appearance of short-period internal waves in more detail, a continuous spectrum or spectrogram was built. To calculate the spectrogram, we used the record of vertical displacement of the thermocline registered by the LTS at station No. 1 (data are shown in Figure 6, top). The LTS record clearly shows inertial period fluctuations. The continuous spectrum (Figure 6, bottom)

shows that high-frequency oscillations (periods 2–8 min) appear on the thermocline with a periodicity close to 17 h, and their appearance coincides with the passage of the crests of inertial internal waves. The current spectra constructed from the records of the line temperature sensors at stations 2 and 3 look similar and confirm the appearance of packets of short-period waves when the crests of inertial waves pass.

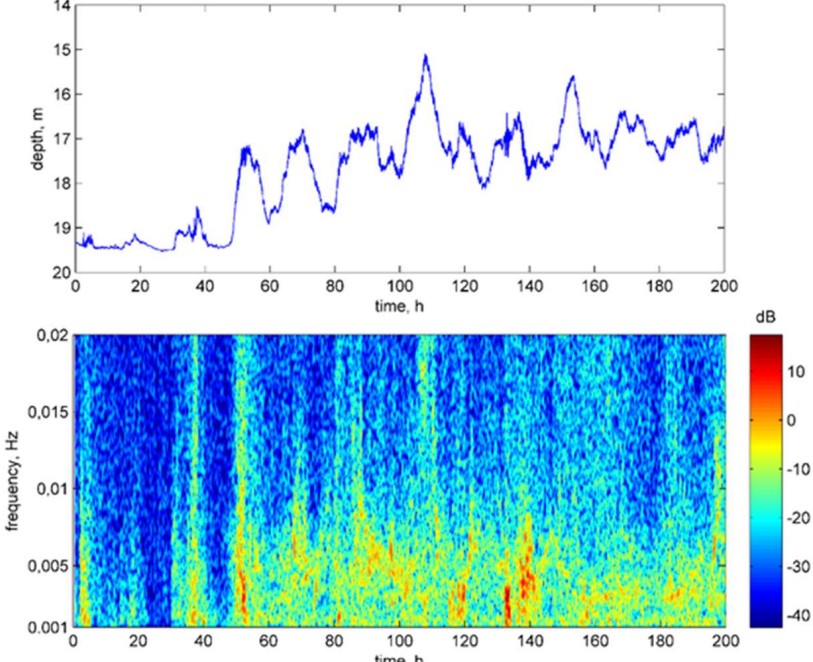

**Figure 6.** Temperature record of LTS No.1 for the entire period (**top**) and the spectrogram calculated from it (**bottom**), showing the inertial periodicity of the appearance of trains of short-period internal waves and their binding to the crests of inertial internal waves.

A more detailed examination of the LTS records confirmed the binding of packets of short-period internal waves to the crests of inertial internal waves. Figure 7 depicts a record of an internal inertial wave made by an LTS, which clearly shows the appearance of short-period oscillations on its crest. It should be noted also that the leading edge of the long internal wave is steeper than the trailing edge. This feature indicates a non-linear transformation of an internal wave approaching the coast.

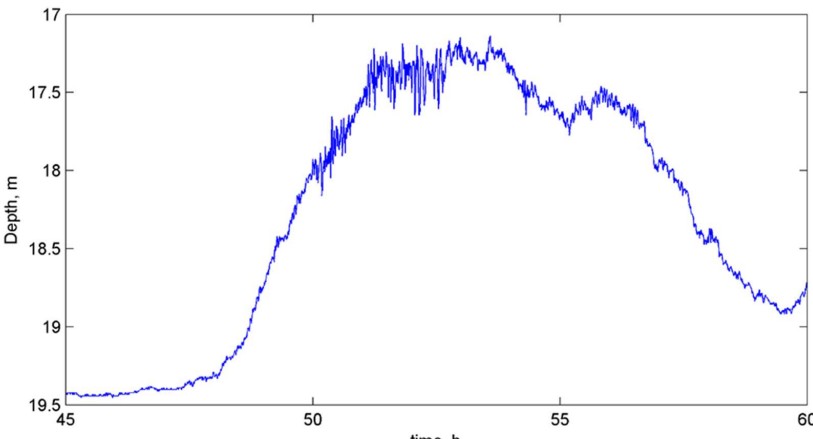

**Figure 7.** Profile of one of the inertial internal waves recorded by the LTS. The appearance of a train of short-period internal waves on the crest of a long wave is clearly visible.

A fragment of a wave train on the crest of an inertial wave is shown separately in Figure 8. The train consists of 10 waves with periods from 4 to 8 min, and wave heights from 0.4 m to 1 m. The parameters of these waves are typical for short-period internal waves recorded in the Black Sea.

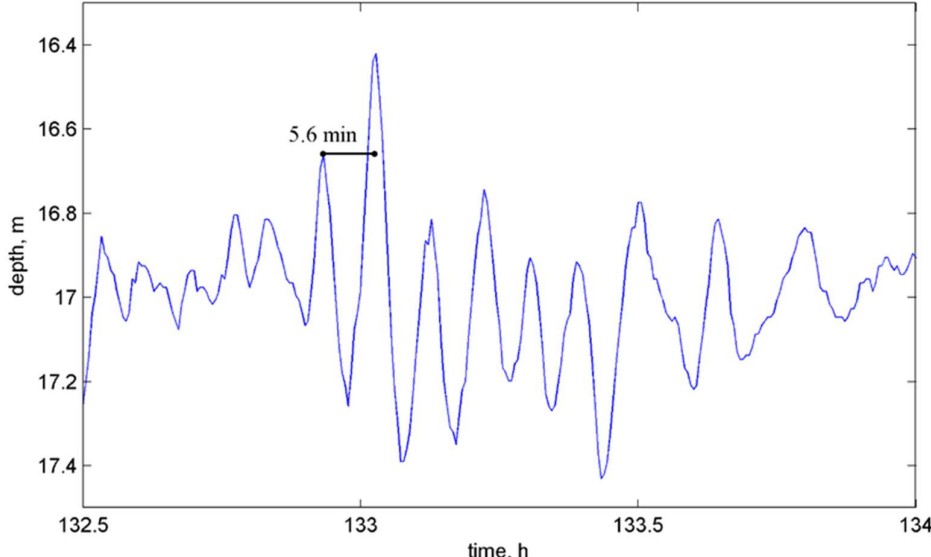

**Figure 8.** LTS record of a train of short-period internal waves on the crest of an inertial wave.

Oscillations of the water column with an inertial period were moved synchronously for all the layers from the surface to the bottom, indicating their belonging to the first mode. The same has been found for short-period waves, except for one time interval belonging to the beginning of an observation (for 24 October from 19:59:30 to 21:59:30, Figure 9). At this time, an unusual phenomenon of a sharp expansion of the thermocline was observed. In this case, the isotherm of 11.5 °C shifted down from the horizon of 37 m to 42 m, while the isotherm of 16 °C moved up from 33 m to 28 m. During the initial phase of this phenomenon, the appearance of a solitary wave of the second mode with a period of 8 min was noted, followed by two trains of short waves of the first mode, with periods of about 4 min. The height of the second mode wave was about 4 m, while the heights of the first mode waves were in the range of 0.5–4 m. For mode 1 waves, the height was determined as the range of oscillations of isotherms from the trough to the crest within the wave period. For mode 2, the wave height is determined similarly only in the section of the water column, where the movements of the layers are in phases.

*The Definition of the Wave's Directions and Phase Speed of Short-Period Internal Waves*

Using a spatial antenna of three LTSs, spaced apart at the corners of a triangle with sides of several tens of meters, we were able to determine the directions of propagation of passing short-period internal waves (angles of arrival of waves; azimuths), and also determine their phase speeds. It is supposed that the fronts of the internal waves are rectilinear, and based on this, we compared the wave delays between the three sensors through which the trains passed. The rectilinear of the short-period wave fronts on a scale of several hundred meters in the coastal zone of the Black Sea was confirmed repeatedly by the results of remote sensing, and is beyond doubt (see, for example, Lavrova et al. [24]). The azimuths and phase speeds were determined by a standard method based on the definition of time delays of signal arrivals between pairs of receivers located in the corners of a triangle. This method works well with a plane wavefront model, and a good signal-to-noise ratio. First, the signals were filtered in the band 0.002–0.00333 Hz (periods of 300–500 s). Then, using cross-correlation analysis, the delays of signal arrival to pairs of receivers were determined. After that, the azimuths and phase speeds were measured. Figure 10 presents the analysis results. The estimated speeds of the propagation of the majority of short-period internal waves were in the range from 0.1 m/s to 0.4 m/s. The obtained

azimuths of wave arrival showed a picture of the almost constant presence of two short-period internal wave systems in the coastal zone. Some part of the waves moves from the open sea to the shore, and others from the coast in the opposite direction. More information is presented in Figure 11, where the histograms of the azimuths and phase velocities for the entire observation period are displayed.

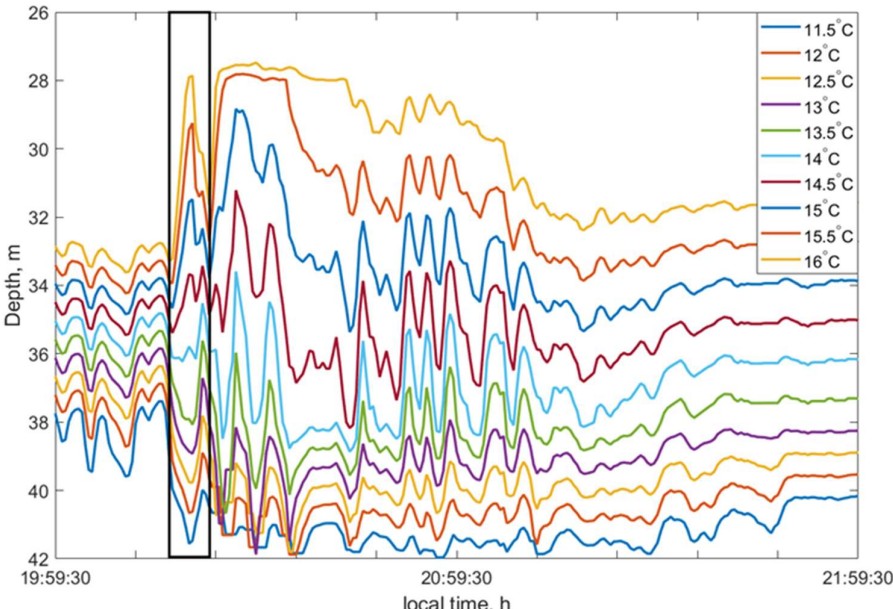

**Figure 9.** The phenomenon of a sharp expansion of the thermocline on 24 October, with the appearance of a leading solitary internal wave of mode 2 (highlighted by a rectangle), and subsequent trains of waves of mode 1.

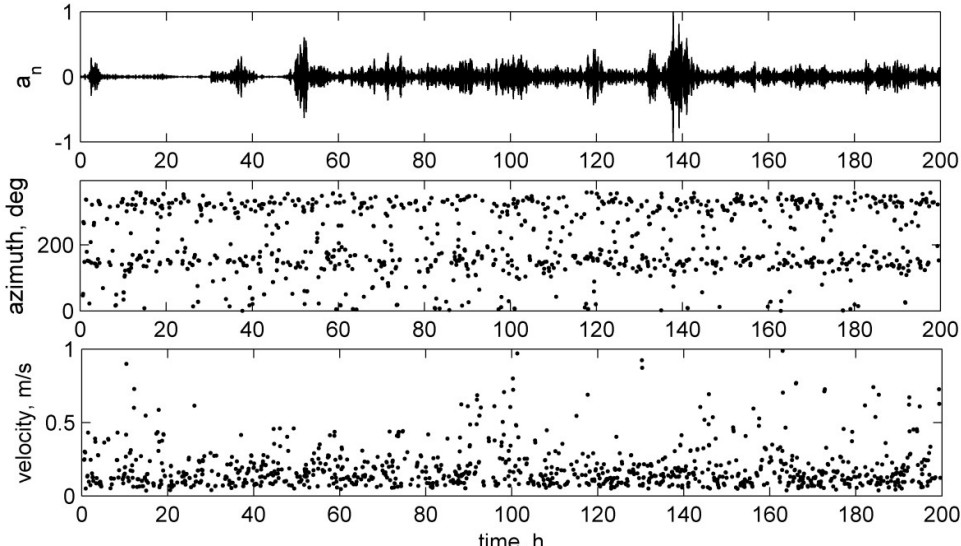

**Figure 10.** The filtered normalized LTS record in the band 0.002–0.00333 Hz (periods of 300–500 s). The azimuths of the arrival of the waves are in the middle. The phase speeds of the waves are below.

The maximum of the azimuth histogram of 148° corresponds to the direction from the measurement site near Cape Sukhumsky to Cape Kodor. The water area near Cape Kodor, where the eponymous river flows into the sea, is placed 15 km to the southeast from Cape Sukhumsky. It is a region characterized by the appearance of freshened water plumes with hydrological fronts, which often propagate to the northwest and approach Cape Sukhumsky. Another maximum of the histogram, 333°, is approximately equal to the first plus 185°. This is probably due to the direction of the arrival of internal waves,

which are reflected from the coast. The phase velocities of the waves vary from 0.05 m/s to 0.35 m/s; in this range, the most common values lie within 0.065–0.18 m/s. Based on the measured data on the speeds and periods of the waves, we obtained an approximate range of short-period wavelengths of within 30–100 m.

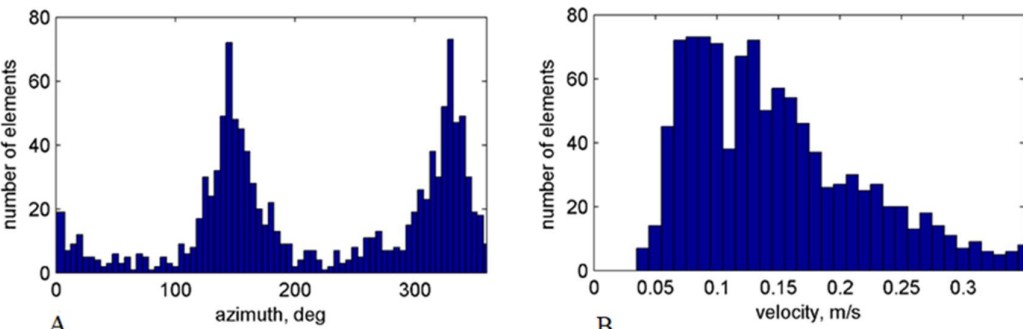

**Figure 11.** (**A**) The histogram of azimuths of the arrival of the waves (interval 5°) and (**B**) the histogram of internal wave phase speeds (interval 0.01 m/s) for the periods of 300–500 s.

## 5. Discussion

Let us compare the observed parameters of internal waves with what the theory predicts. For this, a numerical solution of the Cauchy boundary value problem [35] (with boundary conditions of $w(0) = 0$, $w(\text{н}) = 0$) was found:

$$\frac{d^2w}{dz^2} + \frac{N^2 - \omega^2}{\omega^2 - f^2}k_h^2 w = 0 \tag{1}$$

where $w$ is the amplitude of vertical velocity, $k_h$ is the horizontal wave number, $N$ is the buoyancy frequency, $f$ is the inertial frequency, н is the depth of the sea, and $w$ and $N$ depend on $z$.

To solve this problem, we used the program from V. Goncharov [36], in which the vertical density profile of the medium by layers is set. A more detailed description is given at the end of the article in Appendix A. To solve (1), we used density profiles at the measurement site, which was observed at the beginning of the experiment on 24 October, and again on 30 October 2019.

Figure 12 depicts the dispersion relationship for waves of 1 baroclinic mode (A) and dependence of the phase velocity on the wave vector (B), obtained as a result of solution (1), and also indicates the boundaries of the measured parameters of internal waves (showed by rectangles). A comparison of the ranges of measured parameters of internal waves is shown in Figure 12, with theoretical curves in good agreement.

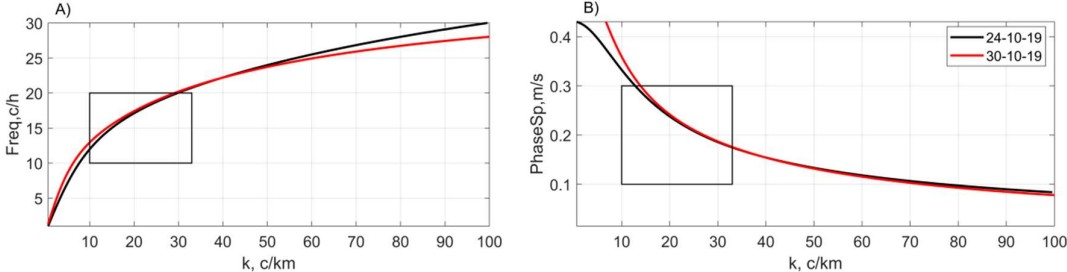

**Figure 12.** The dispersion relationship $f(k)$ (**A**) and dependence of the phase velocity on the wave vector $c(k)$ (**B**) for the internal waves of the first mode, obtained based on the solution of (1).

Let us return to the issue of the direction of the propagation of the observed short-period internal waves. Previously in the study area, we measured short-period internal waves by using only single sensors. Therefore, we knew the typical periods of the waves and their amplitudes, but we did not have information about the wave lengths and the direction of their propagation. Nevertheless, it was

possible for us to use additional information on the propagation of internal waves and their lengths in this region, obtained by remote sensing data. In September 2016, we carried out observations near Cape Sukhumsky and, as a result, the approach of a hydrological front carrying cold waters to the coastal zone was recorded [37]. In this research, it was demonstrated that the approach of the front towards Cape Sukhumsky was associated with the transfer of fresh water of the Kodor river plume. The fresh river plume tracks, moving to the northwest, were accompanied by local hydrological fronts and generated packets of short-period internal waves.

Figure 13 presents the MSI Sentinel 2-A optical range satellite image for the measurement site on 10 September 2017. Due to the color contrast, the plume at the area of the confluence of the river to the sea and tracks of fresh water extending along the coast to the northwest towards Cape Sukhumsky are clearly visible in the image. The water area near Cape Sukhumsky that is marked in the red rectangle is shown in Figure 14 at an enlarged scale. Packets of internal waves are clearly visible on it, moving shoreward in the direction of 350° to the cape. The front border of the train is located at a distance of about 3 km from the coast. It was clear that soon it will approach the place where the spatial antenna of LTS was installed in our observations in 2019. The packet of waves consists of nearly 20 waves with lengths close to 100 m. Thus, the lengths of these waves and their directions coincide with the parameters of the waves measured by the spatial systems of the LTSs. The hydrological conditions when this image was taken were close to those of our observations in 2019. The ADCP data measured in 2019 showed the presence of an alongshore northwestern current during our experiment. Thus, we can conclude that the short internal waves moving towards the coast identified in the analysis, like the waves in the image, owe their origin to the northwestern transport of fresh water from the river Kodor. This is one of the possible sources of the observed trains of short-period internal waves, but there is another. We also noted the appearance of short-period internal waves at the moments when the crests of inertial internal waves approached the coastal zone. The appearance of packets could be explained by the fact that, at these moments, the thermocline is pressed closer to the sea surface, shear currents increase, and the generation conditions of the waves by the mechanism of the Kelvin–Helmholtz shear instability are created.

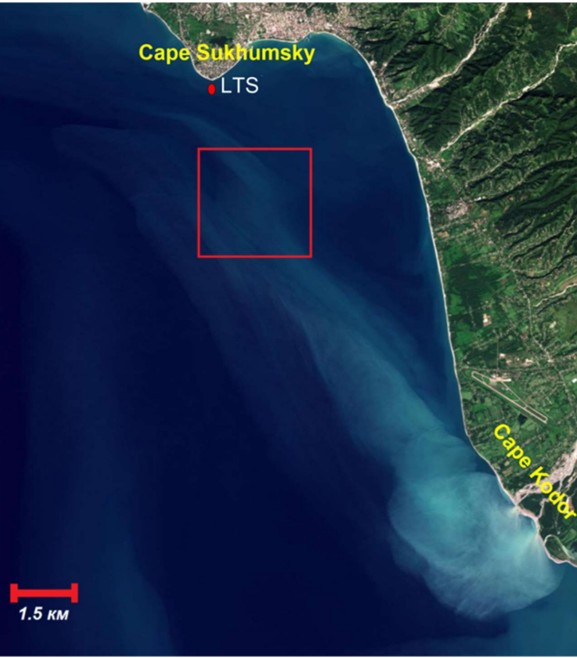

**Figure 13.** MSI Sentinel 2-A optical range satellite image for the measurement site on 10 September 2017, in the Northeast coast of the Black Sea The tracks of fresh water propagate along the shore to the northwest, towards Cape Sukhumsky. The red frame marks the sea area shown in Figure 14 at an enlarged scale.

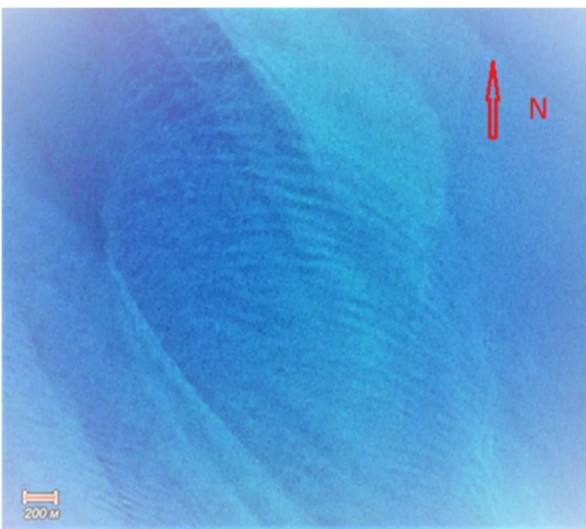

**Figure 14.** Fragment of the image from Figure 13, which shows surface manifestations of short-period internal waves moving shoreward towards Cape Sukhumsky.

The inertial internal waves of the second mode were registered on the Black Sea shelf [38]. These observations were made near the South Coast of Crimea, where the shelf is gentle. In our measurements on a narrow, steep shelf, there were only mode 1 inertial internal waves. As for short-period waves, as previously mentioned, the observed short internal waves belonged to the first mode, with the exception of one case. In the experiment of 2016 [37], the internal waves of the first mode prevailed among short-period waves. However, once a train of waves of the second mode approaching the coast was observed at the moment of a short-term expansion of the thermocline thickness. The situation was similar to our case in 2019 (see Figure 9). At that time, before the appearance of the wave train, which included the second mode wave, the thermocline also expanded. This feature is worthy of attention and requires additional further research.

## 6. Conclusions

Internal waves in the Black Sea differ significantly from their analogs in the ocean. The main difference lies in the lower wave amplitudes, and the fact that the tide—the major source of the generation of internal waves in the ocean—does not work here. Like in the Black Sea, in other closed basins, internal waves are also observed. Depending on the spatial dimensions of water bodies, the field of internal waves has its specifics. Taking into account the abovementioned differences between waves in closed seas and in the ocean, we can say that, nevertheless, in the Caspian Sea [32,39] as well as in the Black Sea, internal waves are similar to oceanic ones. At the same time, in lakes of smaller sizes, edge internal waves play an important role [40,41]. Trains of intense soliton-like waves or internal solitons are observed everywhere. Their appearance is associated with the intense impact of external sources on the stratified environment, such as atmospheric influences, which do not occur often. Our observations, during which we saw regular approaches of inertial internal waves, as well as the appearance of trains of short-period waves of relatively low amplitude, represent the typical and prevailing conditions of the Black Sea shelves at this time of the year.

For many years, a vast amount of information about internal waves on the Black Sea shelf has been collected. However, in most studies, these data were concerned with the internal wave field of ordinary (gentle and wide) shelves. This paper presents the results of the first and most detailed experiment measuring the field of internal waves in the coastal zone of the narrow and steep shelf of the Black Sea, carried out in recent years near Cape Sukhumsky. In this study area, the deep waters are located close to the coast, which imposes its features on the nature of the internal waves. The main feature of the experiment was a system of spatial temperature sensors used to measure the parameters of the internal

waves, which made it possible to obtain such important wave parameters as direction and speed of propagation. The obtained data showed that the internal wave field of this region consists of two main components: long-period internal waves close to local inertial ones, and short-period waves of the first baroclinic mode. The presence of these two ranges of internal waves was conclusively indicated by the peaks of the frequency spectra of the vertical displacements of the thermocline, calculated from a 10-day record. Near-inertial waves with a period close to 17 h regularly approach the shore; their amplitudes reached 8 m. Two systems in the short-period wave field were revealed. One is connected with waves moving shoreward, and the other with the reverse direction. The periods of the waves are 2–8 min; the heights are within 0.5–4 m. The wave lengths are from several tens to hundreds of meters. A principal feature revealed was the detection of trains of short-period internal waves moving towards the coast at the moments of the approach of the crests of inertial waves. Their generation at the crests of long internal waves can be associated with the mechanism of the Kelvin–Helmholtz shear instability. Additionally, some of the short-period waves approaching the coast owe their origin to local fronts formed from the plumes of fresh water of the river Kodor, which are transferred by the current to Cape Sukhumsky. An interesting finding was the detection of the frequent presence of short-period internal waves moving from the coast. The origin of these waves could be attributed to reflected waves from the narrow steep shelf. The short internal waves reflected from the coast are probably the main distinguishing feature of the internal wave field of the narrow shelf, compared to the internal waves of an ordinary or wide shelf.

**Author Contributions:** Conceptualization, A.S., O.P.; methodology, A.S., O.P., D.D. and G.K.; software, D.D. and O.P.; data analysis, A.S., O.P. and E.K.; investigation, A.S., E.K., O.P., and G.K.; writing—original draft preparation, A.S., O.P. and E.K. All authors have read and agreed to the published version of the manuscript.

**Funding:** This research was funded by the Russian Foundation for Basic Research (grant number 19-52-40007 and 19-05-00715).

**Acknowledgments:** The work was supported by the Ministry of Education and Science of the Russian Federation (state assignment no. 0149-2019-0011), in terms of developing a marine experiment methodology and data analysis. The authors would like to thank L. Tarasov, D. Belov and V. Chekayda of Andreyev Acoustics Institute for their help in carrying out measurements in the sea. We are grateful to V. Goncharov of Shirshov Institute of Oceanology for providing the code for the numeric modelling. We are grateful to our colleagues from the Institute of Ecology for their help taking continuous measurements from a stationary platform. We thank the anonymous reviewers for their helpful suggestions.

**Conflicts of Interest:** The authors declare no conflict of interest.

**Data Availability:** Experimental data are archived at the Ocean Acoustics Laboratory (Shirshov Institute of Oceanology, Russian Academy of Sciences) and available upon request.

## Appendix A

The calculation of the eigenfunctions and dispersive curves of internal waves was carried out by numerically solving the equation of internal waves using the program elaborated by V.Goncharov [36]. The program solves the following equation for internal waves:

$$\frac{d^2W}{dz^2} - \frac{N^2}{g}\frac{dW}{dz} + k^2\frac{N^2 - \omega^2}{\omega^2 - F^2}W = 0, \ \left.\frac{dW}{dz}\right|_{z=0} = \frac{gk^2}{\omega^2 - F^2}W(0), \ W(-H) = 0$$

To calculate the parameters of internal waves in a horizontally homogeneous layer of a constant depth H of an incompressible fluid (disregarding the Boussinesq approximation), the equilibrium state of the fluid is assumed to be at rest (no flows), as the initial conditions are given a layer-by-layer unperturbed vertical profile of the density of the medium, obtained from field observations. A model of the ocean (sea) is considered in the form of a system of horizontally homogeneous liquid layers with an upper free boundary $z = 0$ and an absolutely rigid bottom at $z = -H$, where $z$ is the vertical coordinate, $\{x, y\}$ = r—horizontal (Figure A1). In each $n$-th layer, for the hydrological parameters of the environment, the Väisälä frequency $N_n$ is assumed to be constant, and the equilibrium water density

$\rho_0(z)$ is continuous at the boundaries of the layers. With a sufficiently large number of layers $n$, such a model of the medium describes the measured parameters of hydrology well.

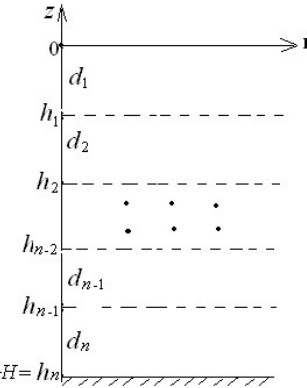

**Figure A1.** Scheme of a layered homogeneous model of a liquid layer [36].

The program is based on an algorithm for the numerical calculation of the dispersion relationship, which uses the impedance method (discrete analogue of the known sweep method), when the ratios of functions ($W'/W$, or $P/Z$) that do not contain exponentially growing members are recalculated. This makes it possible to calculate with sufficient accuracy the mode dispersion $\omega_m(k)$ and the profiles of their eigenfunctions under complex hydrological conditions.

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
