# Peer review of "Internal Waves Study on a Narrow Steep Shelf of the Black Sea Using the Spatial Antenna of Line Temperature Sensors"

_jmse, doi:10.3390/jmse8110833_

Round 1

Reviewer 1 Report

In this paper, the authors report on measurements of internal waves, which they have carried out in the Black Sea using an antenna of line temperature sensors (LTS) together with a thermistor chain and an ADCP. Their records show the presence of near-inertial internal waves with a period close to 17 h and short-period internal waves with periods of 4–8 min. The short-period internal waves are located at the crest of the inertial internal waves. The current spectra also show this phenomenon. These are nice results. However, the paper needs some amendments before publication.

I have the following major suggestions and questions:

  • 1: The authors should add a map (maybe as an inset in Fig. 1) showing the area in the Black Sea in which the measurements were carried out.
  • 2: What is the reason for the strong stratification? As the authors write on page 7, it is associated with fresh water of the Kodor river plume. In this case, the stratification should be due primarily to the difference in salinity. Thus, the study deals with internal waves associated with a river plume and this should be incorporated in the title. The fact that the internal waves are measured close to a steep shelf is of secondary relevance, since there internal waves are reflected, not generated.
  • 5: The figure captions of this figure are too short. What does the first plot show? What shows the insert? What shows the second plot?
  • 6: The figure captions of this figure are incomplete. What are the units in the abscissa? Where is 24 October, 19:59:30 in Figure 6? Is it UTC? The authors should indicate by an arrow the location of the mode- 2 internal wave. The plot is too busy. Delete every second curve.
  • The authors should address the generation mechanism of the internal waves. How are the near-inertial internal waves with a period close to 17 h generated? By a previous strong wind event?
  • How was the buoyancy frequency N determined? Was the salinity measured? The gradient of the density determines N, not the gradient of the temperature.
  • 9, figure captions: What are the red lines?

Minor issues:

Line 73: Write “we have analyzed only the data from…”

Line 11: Write “from the temperature record…. “

Equation 11: Write everywhere “w”, and not “w and W”.

Line 127: Write “inset”.

Line 200: Write” Let us go back…

Line 251: Put a comma after “many years… “

Reviewer 2 Report

See the attached

Reviewer 3 Report

This article presents result from approximately 10 days of measurements of internal waves on the steep, narrow northeastern shelf of the Black Sea using three moorings with line temperature sensors and one mooring with thermistors. The results show the presence of long internal waves near the inertial frequency (~17 h) and very short period (several minutes) internal waves which they associate with the fresh water outflow from the Kodor River. Two major comments that must be addressed before considering the manuscript for publications are:

  1. Complete lack of reference to the very extensive body of literature on international investigations of internal waves.
  2. There is a strong focus on the very short period internal waves which lacks motivation or explanation of the significance or importance. They do not provide compelling evidence that these waves are real and not simply high frequency noise in the spectrum. On the other hand, the near inertial frequency waves receive only a passing mention with no discussion.

Other comments that need to be addressed are listed in order of the manuscript.

Lines 71-73: The ADCP was located in very shallow water, above the thermocline, in the upper mixed layer. How representative are these measurements of the conditions in deeper water where their internal wave sensors were located?

Line 74, Fig. 1: A location map (at least showing the northeastern shelf region of the Black Sea would be helpful).

Lines 89-91. The thermocline shifted upward by 10 m over the 10 day period. This is counterintuitive for this season. In line 98-99 they mention that conditions were sunny and calm. Is this enough to explain the upward shift?

Line 103-108 and Fig 4: The confidence interval for the power spectrum should be shown in the figure. This might also help convince the reader that the “peak” showing the short period waves is no simply high frequency noise.

Lines 123-124. This sentence is not clear. Perhaps they mean to say that the high frequency waves are superimposed on the near inertial frequency waves, appearing mainly near the peaks of the near inertial waves. Also, what is the period of these high frequency waves? In various places in the text they give several ranges: 4-8 min, 6-10 min, and here 2-8 min. Which is it? And what is the buoyance frequency here? It would be helpful to know since this places a limit on the internal wave frequencies.

Line 131. By the “first mode” do they mean the first internal, baroclinic mode?

Line 200. Should be “Let us return to…”

Lines 200-229. One of the major comments above was related to the lack of motivation or explanation of the importance or significance of high frequency internal waves. Here would be the place to discuss this. Also, are there any other documented cases or examples of similar high frequency (near the buoyancy frequency) waves. This question would also be resolved if the authors do a wider literature search.

Line 221-223. Following comment on line 71-73 above, how relevant are these current measurements? Perhaps some of the current measurements should be shown.

Lines 231-238 and Figs. 9 and 10. These figures show surface manifestations of internal waves. Do they have any estimate of the surface wave heights? Generally we expect the relationship between the surface response to internal waves to be on the order of the density difference ratio (as in reduced gravity). Perhaps some discussion of this is in order here. Do these surface manifestations correlate with the near inertial internal waves or the high frequency internal waves. This must be clarified.

Reviewer 4 Report

Review for the manuscript

“Internal Waves Study on Narrow Steep Shelf of the Black Sea Using Spatial Antenna of Line Temperature Sensors” by Andrey Serebryany, Elizaveta Khimchenko, Oleg Popov, Dmitriy Denisov and Genrikh Kenigsberger

The manuscript describes the measurements of internal waves on the northeastern shelf of the Black Sea and the analysis of the resulting records. The presented study is interesting and relevant, it is quite consistent with the topics of the journal “Journal of Marine Science and Engineering”.

My general comments are as follows:

  1. The manuscript requires moderate editing of English language and style.
  2. The term “Narrow Steep Shelf” used in the title of the manuscript and everywhere throughout the text requires clarification and explanation, specifying typical horizontal and vertical spatial scales compared to a regular (“ordinary”) shelf.
  3. The manuscript lacks a comparison with observational data in other regions of the World Ocean. Perhaps, somewhere similar processes were observed in similar conditions. Since we are talking about internal waves in a micro-tidal sea, it also suggests a comparison with physical processes in lakes, where near-inertial baroclinic waves are also often observed (see, for instance, Valipour, R., Bouffard, D., Boegman, L., & Rao, Y. R. (2015). Near‐inertial waves in Lake Erie. Limnology and Oceanography60(5), 1522-1535), accompanied by smaller-scale disturbances.
  4. In the list of references, 11 out of 14 items belong to the authorship of one or more co-authors of the submitted manuscript. I think such a self-citation is inappropriate.
  5. In my opinion, after the introduction, it is necessary to add a section describing the geographic and hydrological conditions of the experiment area (typical seasonal features, the presence of river runoff, inhomogeneities of the bottom relief, etc.). This information is scattered in different places in the text, but it would be good to describe all this in one place at the beginning of the article, so that the reader could understand the general picture of the ongoing processes and the determining factors.

The following technical points should be corrected and / or clarified in more detail:

  1. In the description of Fig. 2, 3 in the text and in the figure captions it is necessary to add the numbers of stations, which the data presented there correspond to.
  2. When constructing the average spectrum shown in Fig. 4, which physical field was analyzed - temperature or isotherm displacement, at what level or for what temperature value? What range was the averaging over?
  3. How the position of the thermocline was determined, shown in Fig. 5? Which factors explain the great difference in the amplitudes (almost two times) of the thermocline displacements in the records shown in Fig. 3 and 5?
  4. How the heights of the first and second mode internal waves were determined?
  5. When describing the statistical properties of empirical distributions of wave directions and velocities, it is better to use standard well-known parameters (mode, median, etc.) instead of “the most common values”.
  6. The description of the eigenvalue problem (1) (lines 188-190) is given disorderly and should be rewritten in a proper way with an indication of the methods for solving ща this rather nontrivial problem and with the necessary links and references. The use of the background temperature profile at the initial moment of recording also raises questions, since, as we can see from Fig. 3, the thermocline has significantly shifted in the remaining days, respectively, the background conditions and wave characteristics have changed.
  7. It is necessary to clarify what is shown by the horizontal and vertical lines in Fig. 9, or remove these lines.

I recommend accepting the article after revision.

Round 2

Reviewer 3 Report

In my original review, in lines 89-91 I asked about the upward shift of the thermocline by about 10 m noting that this was counter intuitive for this season (Oct-Nov). The authors' response was "We disagree with this. Recall that the measurements took place in late October and early November. At this time, even in clear weather, the cooling process begins." This response does not answer my comments. If anything it further strengthens my comment since they too know that the "cooling process begins". Therefore my comment still stands and I would like to see some explanation for the upwards shift of the thermocline at time when would expect cooling to lead to an erosion or downward shift of the thermocline.

Author Response

Our reply is in the file. 
